# Is Adversarial Training Really a Silver Bullet for Mitigating Data Poisoning?

**Rui Wen[1], Zhengyu Zhao[1], Zhuoran Liu[2], Michael Backes[1], Tianhao Wang[3], Yang Zhang[1]**
[1]CISPA Helmholtz Center for Information Security, [2]Radboud University, [3]University of Virginia
`{rui.wen,zhengyu.zhao,director,zhang}@cispa.de,`
`z.liu@cs.ru.nl, tianhao@virginia.edu`

## Abstract

Indiscriminate data poisoning can decrease the clean test accuracy of a deep learning model by slightly perturbing its training samples. There is a consensus that such poisons can hardly harm adversarially-trained (AT) models when the adversarial training budget is no less than the poison budget, i.e., $\epsilon_{\text{adv}} \geq \epsilon_{\text{poi}}$. This consensus, however, is challenged in this paper based on our new attack strategy that induces *entangled features* (EntF). The existence of entangled features makes the poisoned data become less useful for training a model, no matter if AT is applied or not. We demonstrate that for attacking a CIFAR-10 AT model under a reasonable setting with $\epsilon_{\text{adv}} = \epsilon_{\text{poi}} = 8/255$, our EntF yields an accuracy drop of $13.31\%$, which is $7\times$ better than existing methods and equal to discarding $83\%$ training data. We further show the generalizability of EntF to more challenging settings, e.g., higher AT budgets, partial poisoning, unseen model architectures, and stronger (ensemble or adaptive) defenses. We finally provide new insights into the distinct roles of non-robust vs. robust features in poisoning standard vs. AT models and demonstrate the possibility of using a hybrid attack to poison standard and AT models simultaneously. Our code is available at `https://github.com/WenRuiUSTC/EntF`.

## 1 Introduction

Indiscriminate data poisoning aims to degrade the overall prediction performance of a machine learning model at test time by manipulating its training data. It has been increasingly important to understand indiscriminate data poisoning as web scraping becomes a common approach to obtaining large-scale data for training advanced models (Brown et al., 2020; Dosovitskiy et al., 2021). Although slightly perturbing training samples has been shown to effectively poison deep learning models, there is a consensus that such poisons can hardly harm an adversarially-trained model when the perturbation budget in adversarial training, $\epsilon_{\text{adv}}$, is no less than the poison budget, $\epsilon_{\text{poi}}$, i.e., $\epsilon_{\text{adv}} \geq \epsilon_{\text{poi}}$ (Fowl et al., 2021a;b; Huang et al., 2021; Tao et al., 2021; Wang et al., 2021; Fu et al., 2022; Tao et al., 2022). In particular, Tao et al. (2021) have proved that in this setting, adversarial training can serve as a principled defense against existing poisoning methods.

However, in this paper, we challenge this consensus by rethinking data poisoning from a fundamentally new perspective. Specifically, we introduce a new poisoning approach that entangles the features of training samples from different classes. In this way, the entangled samples would hardly contribute to model training no matter whether adversarial training is applied or not, causing substantial performance degradation of models. Different from our attack strategy, existing methods commonly inject perturbations as shortcuts, as pointed out by Yu et al. (2022). This ensures that the model wrongly learns the shortcuts rather than the clean features, leading to low test accuracy (on clean samples) (Segura et al., 2022a; Evtimov et al., 2021; Yu et al., 2022).

Figure 1 illustrates the working mechanism of our new poisoning approach, with a comparison to a reverse operation that instead aims to eliminate entangled features. Our new approach is also inspired by the conventional, noisy label-based poisoning approach (Biggio et al., 2012; 2011; Muñoz-González et al., 2017), where *entangled labels* are introduced by directly flipping labels (e.g., assigning a "dog" (or "cat") label to both the "dog" and "cat" images) under a strong assumption that

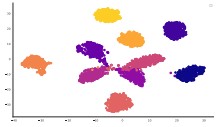 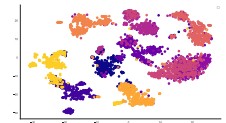 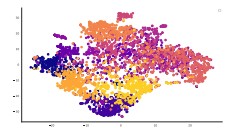 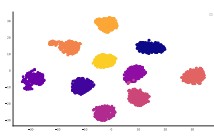

(a) Test Acc: 84.88%          (b) Test Acc: 72.99% ↓          (c) Test Acc: 71.57% ↓          (d) Test Acc: 88.72% ↑

Figure 1: The t-SNE feature visualizations for (a) clean CIFAR-10 vs. poisoned CIFAR-10 achieved by our (b) EntF-pull and (c) EntF-push, which aim to induce entangled features. As a comparison, (d) uses a reverse objective of EntF-push and instead increases the model accuracy. Different from our EntF, existing methods lead to well-separable features (see Appendix A). All t-SNE visualizations in this paper are obtained from the same clean reference model with $\epsilon_{\mathrm{ref}} = 4$.

the labeling process of the target model can be manipulated. However, due to the imperceptibility constraint in the common clean-label setting, we instead propose to introduce *entangled features* represented in the latent space. Our work mainly makes three contributions:

- We demonstrate that, contrary to the consensus view, indiscriminate data poisoning *can* actually decrease the clean test accuracy of adversarially-trained (AT) models to a substantial extent. Specifically, we propose EntF, a new poisoning approach that is based on inducing entangled features in the latent space of a pre-trained reference model.

- We conduct extensive experiments to demonstrate the effectiveness of EntF against AT in the reasonable setting with $\epsilon_{\mathrm{adv}} = \epsilon_{\mathrm{poi}}$ and also its generalizability to a variety of more challenging settings, such as AT with higher budgets, partial poisoning, unseen model architectures, and stronger (ensemble or adaptive) defenses.

- We further highlight the distinct roles of non-robust vs. robust features in compromising standard vs. AT models and also propose hybrid attacks that are effective even when the defender is free to adjust their AT budget $\epsilon_{\mathrm{adv}}$.

## 2   RELATED WORK

**Data poisoning.** Data poisoning aims to compromise a model's performance at test time by manipulating its training data. Related work on poisoning DNNs has mainly investigated targeted, backdoor, and indiscriminate poisoning. Different from backdoor (Gu et al., 2017; Liu et al., 2018; Salem et al., 2022) and targeted poisoning (Muñoz-González et al., 2017; Shafahi et al., 2018; Geiping et al., 2021), which aim to degrade the model on specific (targeted) test samples, indiscriminate poisoning aims at arbitrary clean test samples. Traditional indiscriminate poisoning is based on injecting noisy labels (Biggio et al., 2012; 2011; Muñoz-González et al., 2017); however, they can be easily detected (Shafahi et al., 2018; Song et al., 2022). Recent methods instead pursue "clean-label" poisons by adding imperceptible perturbations. These methods mainly use the error-minimization (Huang et al., 2021; Tao et al., 2021; Fu et al., 2022) or error-maximization loss (Fowl et al., 2021b), with a pre-trained (Fowl et al., 2021b; Wang et al., 2021; Tao et al., 2021) or trained-from-scratch (Huang et al., 2021; Fu et al., 2022) reference model. However, these methods are known to be vulnerable to adversarial training (AT). In particular, two concurrent methods, ADVIN (Wang et al., 2021) and REM (Fu et al., 2022), also attempt to poison AT models, but under easy settings with $\epsilon_{\mathrm{poi}} \geq 2\epsilon_{\mathrm{adv}}$. Generating poisons using a feature-space loss has also been explored (Shafahi et al., 2018; Zhu et al., 2019; Geiping et al., 2021), but without considering AT and in the field of targeted poisoning.

**Adversarial training.** Adversarial training (AT) was recognized as the only promising solution so far to provide robustness against (test-time) adversarial examples (Athalye et al., 2018; Tramèr et al., 2020). It was also recently proved to be a principled defense against indiscriminate poisoning (Tao et al., 2021). The general idea of AT is to simply augment training data with adversarial examples generated in each training step. The single-step approach, FGSM, was initially used by the seminal work of Goodfellow et al. (2015) but has been found to be ineffective against multi-step attacks (Tramèr et al., 2017; Kurakin et al., 2017). To address this limitation, Madry et al. (2018) have proposed the PGD-based AT, which uses the multi-step optimization to further enhance the robustness. Other state-of-the-art methods have been focused on improving this PGD-based AT by,

for example, training on both clean and adversarial examples (Zhang et al., 2019), incorporating an explicit differentiation of misclassified examples (Wang et al., 2020), identifying a bag of training tricks (Pang et al., 2021), or accelerating the training process via gradient recycling (Shafahi et al., 2019). In this paper, we consider three different state-of-the-art AT techniques that adopt the standard PGD and also varied adversarial training budget $\epsilon_{\mathrm{adv}}$.

## 3 ENTANGLED FEATURES (ENTF) FOR POISONING AT MODELS

### 3.1 PROBLEM STATEMENT

We formulate the problem in the context of image classification DNNs. There are two parties involved, the *poisoner* and the *victim*. The poisoner has full access to the clean training dataset $\mathcal{D}_c = \{(\boldsymbol{x}_i, y_i)\}_{i=1}^n$ and is able to add perturbations $\boldsymbol{\delta}^{\mathrm{poi}}$ to each sample and release the poisoned version $\mathcal{D}_p = \{(\boldsymbol{x}_i', y_i)\}_{i=1}^n$, where $\boldsymbol{x}_i' = \boldsymbol{x}_i + \boldsymbol{\delta}_i^{\mathrm{poi}}$. Once the poisoned dataset is generated and released, the poisoner cannot further modify the dataset. Moreover, the poisoner has no control over the target model's training process and the labeling function of the victim. The victim only has access to the poisoned dataset and aims to train a well-generalized model using this dataset. As the victim is aware that the obtained dataset may be poisoned, they decide to deploy adversarial training to secure their model. **The goal of the poisoner is to decrease the clean test accuracy of the adversarially-trained model by poisoning its training dataset.**

When perturbing the clean dataset, the poisoner wants to ensure that the perturbation is imperceptible and can escape any detection from the victim. To this end, the poisoner constrains the generated perturbations $\boldsymbol{\delta}^{\mathrm{poi}}$ by a certain *poison budget* $\epsilon_{\mathrm{poi}}$, i.e., $\|\boldsymbol{\delta}^{\mathrm{poi}}\|_\infty \le \epsilon_{\mathrm{poi}}$. Take the widely-adopted adversarial training framework (Madry et al., 2018) as an example, the victim trains a target model $F$ on the poisoned dataset $\mathcal{D}_p$ by a certain *adversarial training budget* $\epsilon_{\mathrm{adv}}$ following the objective:

$$\arg\min_\theta \ \mathbb{E}_{(\boldsymbol{x}', y) \sim \mathcal{D}_p} \left[ \max_{\boldsymbol{\delta}^{\mathrm{adv}}} \mathcal{L}(F(\boldsymbol{x}' + \boldsymbol{\delta}^{\mathrm{adv}}), y) \right] \ \text{s.t.} \ \|\boldsymbol{\delta}^{\mathrm{adv}}\|_\infty \le \epsilon_{\mathrm{adv}}, \tag{1}$$

where $\boldsymbol{x}'$ denotes the poisoned input, $\boldsymbol{\delta}^{\mathrm{adv}}$ denotes the adversarial perturbations, $\theta$ denotes the model parameters, and $\mathcal{L}$ is the classification loss (e.g., the commonly used cross-entropy loss).

In this paper, we focus on the reasonable setting with $\epsilon_{\mathrm{poi}} \le \epsilon_{\mathrm{adv}}$. In contrast, the two concurrent studies, ADVIN (Wang et al., 2021) and REM (Fu et al., 2022), focus on much easier settings with $\epsilon_{\mathrm{poi}} \ge 2\epsilon_{\mathrm{adv}}$, in which it is not surprising that AT would fail because the clean samples are already out of the $\epsilon_{\mathrm{adv}}$-ball of the poisoned samples (Tao et al., 2021).

### 3.2 METHODOLOGY

In this section, we introduce EntF, our new poisoning approach to compromising adversarial training. The key intuition of EntF is to cause samples from different classes to share entangled features and then become useless for model training, including adversarial training. Specifically, we propose two different variants of EntF, namely EntF-push and EntF-pull. For EntF-push, all training samples in each of the original classes $y$ are pushed away from the corresponding class centroid $\boldsymbol{\mu}_y$ in the latent feature space (i.e., the output of the penultimate layer $F_{L-1}^*$) of a reference model $F^*$, which has totally $L$ layers. The objective function can be formulated as:

$$\mathcal{L}_{\mathrm{push}} = \max_{\boldsymbol{\delta}^{\mathrm{poi}}} \|F_{L-1}^*(\boldsymbol{x} + \boldsymbol{\delta}^{\mathrm{poi}}) - \boldsymbol{\mu}_y\|_2 \ \text{s.t.} \ \|\boldsymbol{\delta}^{\mathrm{poi}}\|_\infty \le \epsilon_{\mathrm{poi}}. \tag{2}$$

For EntF-pull, each training sample is pulled towards the centroid of its nearest class $y'$:

$$\mathcal{L}_{\mathrm{pull}} = \min_{\boldsymbol{\delta}^{\mathrm{poi}}} \|F_{L-1}^*(\boldsymbol{x} + \boldsymbol{\delta}^{\mathrm{poi}}) - \boldsymbol{\mu}_{y'}\|_2 \ \text{s.t.} \ \|\boldsymbol{\delta}^{\mathrm{poi}}\|_\infty \le \epsilon_{\mathrm{poi}}. \tag{3}$$

The above class centroid is computed as the average features of all clean samples $\mathcal{X}$ in that class:

$$\boldsymbol{\mu} = \frac{1}{|\mathcal{X}|} \sum_{\boldsymbol{x} \in \mathcal{X}} F_{L-1}^*(\boldsymbol{x}). \tag{4}$$

We find this simple, average-based method works well in our case, and we leave the exploration of other, metric learning methods (Kaya and Bilge, 2019) to future work.

In order to learn a similar representation space to that of an AT target model, the reference model $F^*$ is also adversarially trained with a certain perturbation budget $\epsilon_{\mathrm{ref}}$. We discuss the impact of $\epsilon_{\mathrm{ref}}$ on the poisoning performance in Section 5. Following the common practice, we adopt the Projected Gradient Descent (PGD) (Madry et al., 2018) to solve the above poison optimization.

**Why adversarial training can be compromised.** Tao et al. (2021) have proved that adversarial training can serve as a principled defense against data poisoning based on the following theorem.

**Theorem 1** *Given a classifier $f : \mathcal{X} \to \mathcal{Y}$, for any data distribution $\mathcal{D}$ and any perturbed distribution $\hat{\mathcal{D}}$ such that $\hat{\mathcal{D}} \in \mathcal{B}_{W_\infty}(\mathcal{D}, \epsilon)$, we have*

$$\mathcal{R}_{\mathrm{nat}}(f, \mathcal{D}) \leq \max_{\mathcal{D}' \in \mathcal{B}_{W_\infty}(\hat{\mathcal{D}}, \epsilon)} \mathcal{R}_{\mathrm{nat}}(f, \mathcal{D}') = \mathcal{R}_{\mathrm{adv}}(f, \hat{\mathcal{D}}).$$

where $\mathcal{R}_{\mathrm{nat}}$ denotes natural risk and $\mathcal{R}_{\mathrm{adv}}$ denotes adversarial risk. Detailed definitions can be found in Appendix B. Theorem 1 guarantees that adversarial training on the poisoned data distribution $\hat{\mathcal{D}}$ optimizes an upper bound of natural risk on the original data distribution $\mathcal{D}$ if $\hat{\mathcal{D}}$ is within the $\infty$-Wasserstein ball of $\mathcal{D}$ (Tao et al., 2021). That is to say, achieving a low natural risk on $\mathcal{D}$ (i.e., high clean test accuracy) requires a low adversarial risk on $\hat{\mathcal{D}}$. This guarantee is based on an *implicit* assumption that adversarial training is capable of minimizing the adversarial risk on the poisoned data distribution $\hat{\mathcal{D}}$, which holds for existing attacks. However, for our EntF, the poisoned data that share entangled features become not useful even for adversarial training, and as a result, the assumption required for the proof is broken. Our experimental results in Appendix B validate this claim.

**Important note on the cross-entropy loss.** The key novelty of our EntF over existing methods lies in not only the attack strategy (entangled features vs. shortcuts) but also the specific loss (feature-level vs. output-level). Existing methods on poisoning standard models have commonly adopted the cross-entropy (CE) loss and concluded that the targeted optimization, either with an incorrect (Fowl et al., 2021b; Tao et al., 2021; Wang et al., 2021) or original (Huang et al., 2021) class as the target, is generally stronger than the untargeted CE. This conclusion somewhat leads to the fact that the two concurrent studies on poisoning AT models (i.e., ADVIN (Wang et al., 2021) and REM (Fu et al., 2022)) have completely ignored the untargeted CE as their baseline.

However, we find that the above conclusion does not hold for poisoning AT models. Specifically, we notice that the untargeted CE can also lead to entangled features to some extent and as a result yield a substantial accuracy drop (12.02%), while its targeted counterpart (i.e, ADVIN (Wang et al., 2021) shown in our Table 1) completely fails. Note that the untargeted CE still performs worse than our EntF-push, especially in the more complex tasks, i.e., CIFAR-100 and TinyImageNet (see Appendix C for details). This indicates that using the CE loss is not an ultimate solution in practice.

## 4 EXPERIMENTS

In this section, we conduct extensive experiments to validate that adversarial training (AT) can be compromised by our new poisoning attack. In particular, we consider challenging scenarios with high AT budgets, partial poisoning, unseen model architectures, or strong (ensemble or adaptive) defenses.

### 4.1 EXPERIMENTAL SETTINGS

We use three image classification benchmark datasets: CIFAR-10 (CIF), CIFAR-100 (CIF), and TinyImageNet (Tin). These datasets have been commonly used in the poisoning literature. We adopt the perturbation budget $\epsilon_{\mathrm{ref}} = 4/255$ for adversarially training the reference model and find that other values also work well (see results in Section 5). PGD-300 with a step size of $0.4/255$ and differentiable data augmentation (Fowl et al., 2021b) is used for poison optimization. If not explicitly mentioned, we focus on the reasonable setting with $\epsilon_{\mathrm{poi}} = \epsilon_{\mathrm{adv}} = 8/255$ and adopt ResNet-18 for both the reference and target models. Additional experimental settings can be found in Appendix D. All experiments are performed on an NVIDIA DGX-A100 server.

Table 1: Comparison of different poisoning methods against adversarial training. ADVIN, Hypocritical[+], and REM also adopt an adversarially-trained reference model, as our methods.

| POISON METHOD | CLEAN TEST ACCURACY ($\%, \downarrow$) |
|---|---|
| NONE (CLEAN) | 84.88 |
| HYPOCRITICAL (TAO ET AL., 2021) | 84.96 |
| UNLEARNABLE (HUANG ET AL., 2021) | 84.91 |
| CLASS-WISE RANDOM NOISE | 84.06 |
| ADVPOISON (FOWL ET AL., 2021B) | 83.11 |
| ADVIN (WANG ET AL., 2021) | 86.76 |
| HYPOCRITICAL[+] (TAO ET AL., 2022) | 86.56 |
| REM (FU ET AL., 2022) | 84.21 |
| ENTF-PULL (OURS) | 72.99 |
| ENTF-PUSH (OURS) | **71.57** |

## 4.2 ENTF COMPARED TO EXISTING ATTACKS UNDER $\epsilon_{\mathrm{adv}} = \epsilon_{\mathrm{poi}}$

We first evaluate the performance of different state-of-the-art poisoning methods against adversarial training in the basic setting with $\epsilon_{\mathrm{poi}} = \epsilon_{\mathrm{adv}}$ on CIFAR-10. As can be seen from Table 1, all existing methods can hardly decrease the model accuracy. Specifically, although ADVIN (Wang et al., 2021) and REM (Fu et al., 2022) have claimed effectiveness in the unreasonable settings with $\epsilon_{\mathrm{poi}} \geq 2\epsilon_{\mathrm{adv}}$, they fail in our reasonable setting. In some cases, the poisons may even slightly increase the model accuracy, which is also noticed in the concurrent work (Tao et al., 2022).

In contrast to existing methods, both our EntF-push and EntF-pull can substantially decrease the model accuracy. Note that decreasing the model accuracy to $71.57\%$ is dramatic because it equals the performance achieved by directly discarding $83\%$ of the original training data (see more relevant discussions in Section 4.5). In addition, EntF-push and EntF-pull achieve similar results but obviously, EntF-pull is less efficient because it needs to calculate and then rank the distance between each sample and class centroid. Moreover, the class selection strategy in EntF-pull may have an impact on the final performance (see more analysis in Appendix E). For these reasons, if not specifically mentioned, we choose to use EntF-push in the following experiments.

Table 2: Evaluating EntF on different datasets.

| POISON METHOD \ DATASET | CIFAR-10 | CIFAR-100 | TINYIMAGENET |
|---|---|---|---|
| NONE (CLEAN) | 84.88 | 59.50 | 51.95 |
| ENTF (OURS) | 71.57 | 47.29 | 41.32 |

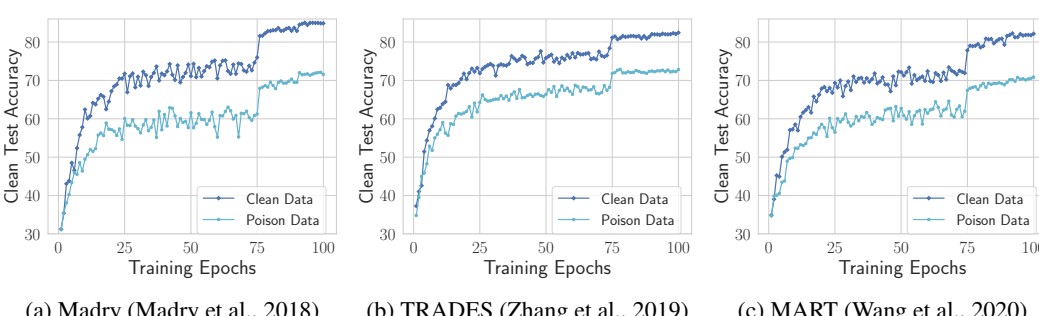

(a) Madry (Madry et al., 2018)    (b) TRADES (Zhang et al., 2019)    (c) MART (Wang et al., 2020)

Figure 2: Evaluating EntF against three different well-known adversarial training frameworks.

## 4.3 ENTF FOR LARGER DATASETS AND OTHER AT FRAMEWORKS UNDER $\epsilon_{\mathrm{adv}} = \epsilon_{\mathrm{poi}}$

The results shown in Table 2 further validate the general effectiveness of our EntF on larger datasets, where the model accuracy consistently drops by more than $10\%$. We also evaluate EntF against

different widely-used AT frameworks. Figure 2 shows the learning curves of the poisoned AT target model that is trained with Madry (Madry et al., 2018), TRADES (Zhang et al., 2019), or MART (Wang et al., 2020). As can be seen, our EntF largely decreases the clean test accuracy in all cases. We can also observe that all three frameworks exhibit a relatively steady learning process, i.e., the model accuracy monotonically increases over epochs, and finally reaches an accuracy that is still lower than that of the model trained on clean data. This pattern is different from that in poisoning standard models, where the model accuracy is found to increase at a few early epochs, and then start to decrease dramatically to the final low accuracy (Huang et al., 2021; Liu et al., 2021; Segura et al., 2022a). This fundamental difference indicates that early stopping cannot be used as an effective defense against poisoning for AT models.

Table 3: Evaluating EntF under different $\epsilon_{\text{poi}}$ vs. $\epsilon_{\text{adv}}$.

| POISON BUDGET \ ADVTRAIN BUDGET | $\epsilon_{\text{adv}} = 4/255$ | $\epsilon_{\text{adv}} = 8/255$ | $\epsilon_{\text{adv}} = 16/255$ |
|---|---|---|---|
| NONE (CLEAN) | 90.31 | 84.88 | 73.78 |
| $\epsilon_{\text{poi}} = 4/255$ | 84.37 | 79.25 | 69.35 |
| $\epsilon_{\text{poi}} = 8/255$ | 75.39 | 71.57 | 63.73 |
| $\epsilon_{\text{poi}} = 16/255$ | 50.27 | 60.29 | 53.03 |

## 4.4 ENTF UNDER HIGHER AT BUDGETS

We further test EntF under higher adversarial training budgets and also consider different poison budgets. As can be seen from Table 3, even with an overwhelming budget, adversarial training can still be largely degraded by our EntF. For example, when the AT budget is $\epsilon_{\text{adv}} = 16/255$, which is $2\times$ larger than the poison budget $\epsilon_{\text{poi}} = 8/255$, our EntF still yields a substantial accuracy drop of 10.05%. In addition, under the same setting with $\epsilon_{\text{adv}} = \epsilon_{\text{poi}}$, a larger poison budget leads to a better poison performance. Specifically, for $\epsilon_{\text{adv}} = \epsilon_{\text{poi}} = 4/255$, model accuracy drops by 5.94% (90.31% $\rightarrow$ 84.37%), while for a larger poison budget, 8/255 or 16/255, the accuracy drops by 13.31% (84.88% $\rightarrow$ 71.57%) or 20.75% (73.78% $\rightarrow$ 53.03%). We can also observe that under a specific poison budget, the model accuracy and AT budget do not have a clear correlation. This is reasonable because although enlarging the AT budget can increase clean accuracy due to higher robustness to poisons, it also inevitably leads to an accuracy drop compared to standard training. We leave detailed explorations for future work.

Table 4: Effects of adjusting the poison proportion. "None (Clean)" shows the baseline results where the rest clean data is used without poisoning.

| POISON METHOD \ POISON PROPORTION | 0.2 | 0.4 | 0.6 | 0.8 |
|---|---|---|---|---|
| NONE (CLEAN) | 83.66 | 81.82 | 79.16 | 73.23 |
| CLEAN+ENTF | 81.41 | 78.67 | 75.84 | 73.56 |
| CLEAN+ENTF (WORSE CASE) | 81.84 | 78.83 | 75.92 | 74.01 |

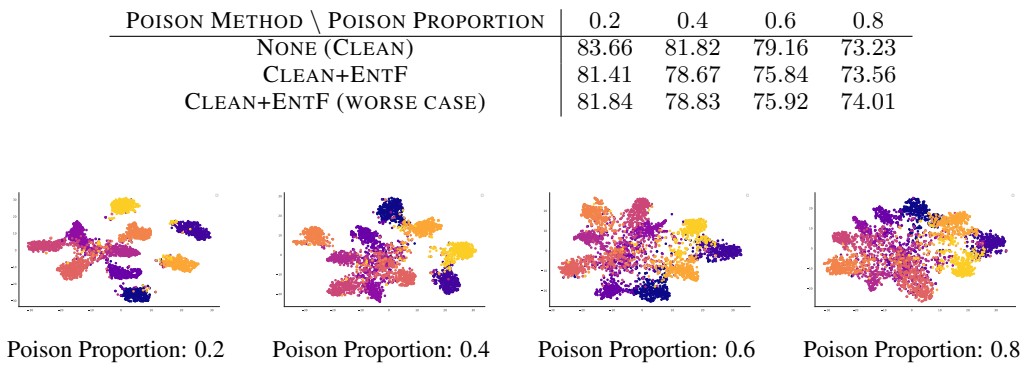

Poison Proportion: 0.2    Poison Proportion: 0.4    Poison Proportion: 0.6    Poison Proportion: 0.8

Figure 3: The t-SNE visualizations for different poison proportions.

## 4.5 ENTF UNDER PARTIAL POISONING

We also examine EntF in a more challenging scenario where only partial training data are allowed to be poisoned. We consider two different poisoning settings where different data are used for calculating the class centroids. Specifically, the first setting is based on the whole original (clean)

dataset but the second, worse-case one is based on only the partial clean data that are allowed to be poisoned. As can be seen from Table 4, our EntF is still effective in this challenging scenario. Recall that other attacks can hardly decrease the model accuracy even when the whole dataset is poisoned. In particular, EntF decreases the model accuracy from $84.88\%$ to $81.41\%$ by only poisoning 0.2 of the training data, and this result is also lower than the baseline that is achieved by directly discarding these poisoned data ($83.66\%$).

We can also observe that the two different poisoning settings yield very similar results. This indicates that the calculation of class centroids in our EntF is not sensitive to the amount of data, and as a result, its efficiency can be potentially improved by using fewer data for the centroid calculation. The fact that poisoning more data yields better performance can also be explained by Figure 3 where a larger poison proportion leads to a larger number of entangled features.

Table 5: Transferability of EntF poisons from ResNet-18 to other model architectures.

| Poison Method \ Target | ResNet-18 | ResNet-34 | VGG-19 | DenseNet-121 | MobileNetV2 |
|---|---|---|---|---|---|
| None (Clean) | 84.88 | 86.58 | 75.99 | 87.22 | 80.11 |
| EntF | 71.57 | 73.05 | 64.66 | 74.35 | 67.21 |

## 4.6 Transferability of EntF to unseen model architectures

The latent feature space that is used for generating poisons is specific to a certain reference model. For this reason, one natural question to ask is whether the poisons generated on one model architecture are still effective when the target model adopts a different architecture. Table 5 demonstrates that the poisoning effects of our EntF optimized against a ResNet-18 reference model can transfer to other target model architectures. Specifically, for the four different (unseen) architectures, the generated poisons are able to degrade the model accuracy to almost the same extent, indicating the strong generalizability of our EntF.

Table 6: Evaluating EntF against defenses that apply both data augmentations and AT.

| Defense | Clean Test Accuracy (%) |
|---|---|
| None (Clean) | 84.88 |
| Adversarial Training | 71.57 |
| +Random Noise | 71.88 |
| +JPEG Compression | 70.40 |
| +Mixup (Zhang et al., 2018) | 71.84 |
| +Cutout (Devries and Taylor, 2017) | 69.81 |
| +Cutmix (Yun et al., 2019) | 68.85 |
| +Grayscale (Liu et al., 2021) | 68.67 |

## 4.7 EntF against other defenses

**Ensemble defenses with data augmentations.** Applying additional data augmentations before standard training has been shown to be able to mitigate the effects of perturbation-based poisons (Huang et al., 2021; Fowl et al., 2021b; Liu et al., 2021; Tao et al., 2021). Here we study if data augmentations can complement adversarial training when facing our EntF. Following previous work, we test a diverse set of data augmentations, including random noise, Mixup (Zhang et al., 2018), Cutmix (Yun et al., 2019), and Cutout (Devries and Taylor, 2017). We also test gray-scale pre-filtering (Liu et al., 2021), which shows strong performance in mitigating unlearnable examples (Huang et al., 2021). Table 6 shows that all the data augmentation methods fail to help AT to mitigate our EntF.

**Adaptive defenses by filtering out entangled samples.** In addition to existing defense methods, we also consider stronger, adaptive defenses that the victim may design based on a certain level of knowledge about our EntF. It is worth noting that, in realistic scenarios, the victim can only leverage a poisoned model, and so it also has no access to a clean AT reference model, which is available to the poisoner, including our EntF. This is reasonable because if it is indeed feasible for the victim to get a clean AT (reference) model, there is no need for dealing with the poisoned data in the first place, and the clean AT model can already be used as an effective target model.

When the victim knows that entangled features have been introduced by EntF, they would filter out the "overlapped samples", which are located close in the feature space but from different classes. We test by removing different proportions of such "overlapped samples" and find that the best setting can only recover the accuracy from 71.57% to 72.43%. We go a step further by considering a stronger victim who even knows the specific algorithm of EntF-push (i.e., Equation 2). In this case, the victim would recover the data by pulling the poisoned samples back toward their original class centroids. We find that this defense is stronger than the above but still can only recover the accuracy to 75.32% (about 10% lower than the clean AT accuracy). We also consider a practical scenario in which the victim can leverage an ST or AT CIFAR-100 model as a general-purpose model. In this scenario, the adaptive defense only recovers the model accuracy to 72.97%.

**Adaptive defenses through feature perturbation-based AT.** We further test a new defense that uses Equation 2 for generating the adversarial examples in AT instead of the common, cross-entropy loss. When trained on clean data, this new AT variant yields an accuracy of 86.84%, similar to that achieved by the conventional AT. However, when trained on our poisoned data, the model accuracy still substantially drops to 72.99%, indicating that it is not a satisfying defense. This defense is even worse than the above one with a pre-trained AT model (72.99% vs. 75.32%). This might be because the class centroids calculated when the model is not well trained (in the early AT training stage) cannot provide meaningful guidance compared to those based on the pre-trained model. As a sanity check, we also try another AT variant that uses a reverse loss of Equation 2 and find that as expected, it causes the model accuracy to drop a lot (to 47.26%).

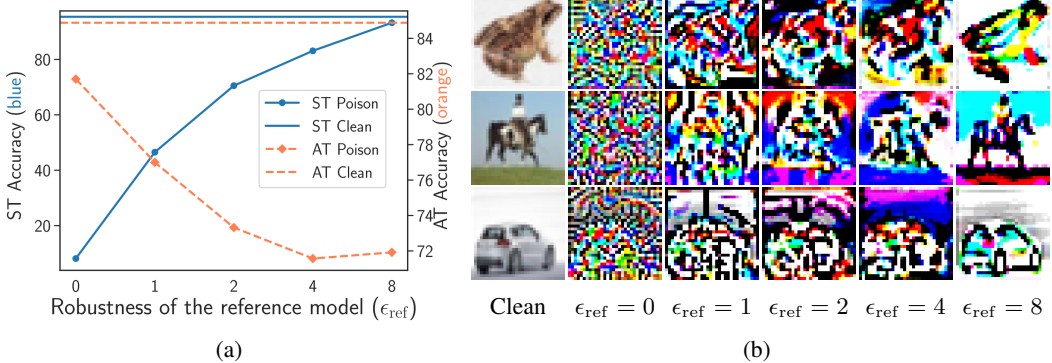

Figure 4: Impact of the robustness of the reference model on poisoning standard (ST, $\epsilon_{\text{ref}} = 0$) vs. adversarially-trained (AT) target model. (a) Clean test accuracy for both ST and AT; (b) Perturbation visualizations for different $\epsilon_{\text{ref}} = 0$. More visualizations can be found in Appendix G.

## 5 POISONING AT VS. ST MODELS

All our experiments have so far been focused on poisoning AT models. However, it is also valuable to study the problem in the context of standard training and figure out the difference. To this end, we analyze the poisons against a standard (ST) model vs. an AT model. Specifically, we adjust the perturbation budget $\epsilon_{\text{ref}}$ for adversarially training the reference model and analyze the poisons in terms of both the poisoning strength and the visual characteristics of perturbations.

As can be seen from Figure 4a, for poisoning the AT model, the poisoning strength is gradually improved as the reference model becomes more robust (i.e., $\epsilon_{\text{ref}}$ is increased). In contrast, for poisoning the ST model, the poisoning is gradually degraded. Concurrent work (Tao et al., 2022) also discusses the impact of $\epsilon_{\text{ref}}$ on poisoning AT models but focuses on a fundamentally different task where the attack aims to degrade the adversarial robustness of AT models (rather than the clean accuracy here). It is also important to note that our EntF can work well under different $\epsilon_{\text{ref}}$ between 2 and 8, where their approach is much more sensitive to the choice of $\epsilon_{\text{ref}}$ in their task (see their Figure 2(a)). We further visualize the perturbations generated with different $\epsilon_{\text{ref}}$ in Figure 4b. As can be seen, when using a standard reference model (i.e., $\epsilon_{\text{ref}} = 0$), the perturbations exhibit noisy patterns, but as the $\epsilon_{\text{ref}}$ is gradually increased, the perturbations tend to be more aligned with image semantics.

These observations suggest that poisoning ST and AT models requires modifying different types of features. More specifically, modifying the robust (semantic) features is the key to poisoning AT

models, while modifying the non-robust features works for ST models. This conclusion also supports the well-known perspective that *non-robust features can be picked up by models during standard training, even in the presence of robust features, while adversarial training tends to utilize robust features* (Ilyas et al., 2019). Figure 4a also confirms that our EntF is effective in poisoning ST models since it can decrease the model accuracy to the random guess level (i.e., 10% for CIFAR-10).

**Hybrid attacks**. The distinct roles of robust and non-robust features in poisoning AT and ST models inspire us to search for a hybrid attack to poison the ST and AT models simultaneously. This is particularly relevant for practical defenders who may want to adjust their AT budget to achieve optimal accuracy. A straightforward idea is to simply average the perturbations generated using an ST reference model and those using an AT reference model. In this case, the poison budget remains unchanged. Alternatively, we propose a hybrid attack that is based on jointly optimizing the perturbations against both ST and AT reference models, balanced by factors $\lambda_i$ that correspond to different AT models. This modifies Equation 2 to the following Equation 5.

$$\mathcal{L}_{\text{hybrid}} = \max_{\boldsymbol{\delta}^{\text{poi}}} \|F^*_{L-1,\text{ST}}(\boldsymbol{x} + \boldsymbol{\delta}^{\text{poi}}) - \boldsymbol{\mu}_{y,\text{ST}}\|_2 + \sum_i \lambda_i \|F^*_{L-1,\text{AT}_i}(\boldsymbol{x} + \boldsymbol{\delta}^{\text{poi}}) - \boldsymbol{\mu}_{y,\text{AT}_i}\|_2. \quad (5)$$

Here we adopt two AT reference models with $\epsilon_{\text{ref}} = 2/255$ and $4/255$. Table 7 shows that both hybrid attacks substantially decrease the model accuracy across different AT budgets $\epsilon_{\text{adv}}$ varying from 0 (i.e., ST) to 16. Specifically, the optimization-based method performs much better than the average-based method. Table 9 in Appendix H further shows the much worse performance of other attacks and confirms the general effectiveness of our hybrid attack strategy, see ours (Hybrid) vs. AdvPoison (Hybrid).

Table 7: Evaluating two hybrid attacks (average/optimization) under different $\epsilon_{\text{poi}}$ vs. $\epsilon_{\text{adv}}$.

| $\epsilon_{\text{poi}} \backslash \epsilon_{\text{adv}}$ | 0/255 | 4/255 | 8/255 | 16/255 | OPTIMAL |
|---|---|---|---|---|---|
| NONE (CLEAN) | 94.59 | 90.31 | 84.88 | 73.78 | 94.59 |
| 4/255 | 51.98/**29.51** | 86.87/**84.48** | 81.24/**80.53** | 71.13/**70.26** | 86.87/**84.48** |
| 8/255 | 22.86/**11.89** | 82.30/**76.55** | 77.19/**74.30** | 66.70/**65.75** | 82.30/**76.55** |
| 16/255 | **5.34**/6.59 | 75.32/**59.55** | 69.41/**66.25** | **59.40**/60.73 | 75.32/**66.25** |

## 6 CONCLUSION

In this paper, we have proposed EntF, a new poisoning approach to decreasing the deep learning classifier's accuracy even when adversarial training is applied. This approach is based on a new attack strategy that makes the features of training samples from different classes become entangled. Extensive experiments demonstrate the effectiveness of EntF against adversarial training in different scenarios, including those with more aggressive AT budgets, unseen model architectures, and adaptive defenses. We also discuss the distinct roles of the robust vs. non-robust features in poisoning standard vs. adversarially-trained models and demonstrate that our hybrid attacks can poison standard and AT models simultaneously.

We encourage future research to analyze EntF in more comprehensive settings and compare it to the current, shortcut-based methods from different angles. In particular, it is important to come up with new defenses against EntF, possibly based on advanced techniques of learning from noisy labels (Song et al., 2022). It is also worth noting that most of the current poisoning studies, including ours, have assumed that the poisoner has access to the training dataset of the target model. This assumption is realistic in specific threat models (e.g., secure dataset release (Fowl et al., 2021a)) but may not be plausible for sensitive/private data. Therefore, it would be promising to extend EntF to addressing data-free poisons (Yu et al., 2022; Segura et al., 2022b).

On the one hand, data poisoning could be potentially leveraged by malicious parties as attacks. In this case, we hope our work can inspire the community to develop stronger defenses based on our comprehensive analysis. On the other hand, when data poisoning is directly used for social good, e.g., for protecting personal data from being misused (Fowl et al., 2021a; Huang et al., 2021; Fu et al., 2022), our new approach for generating stronger poisons leads to stronger protective effects.

ACKNOWLEDGEMENT

We thank all anonymous reviewers for their constructive comments. This work is partially funded by the Helmholtz Association within the project "Trustworthy Federated Data Analytics" (TFDA) (funding number ZTI-OO1 4), by the European Health and Digital Executive Agency (HADEA) within the project "Understanding the individual host response against Hepatitis D Virus to develop a personalized approach for the management of hepatitis D" (D-Solve) (grant agreement number 101057917), and by NSF grant number 2217071.

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

## A    WELL-SEPARABLE FEATURES IN EXISTING METHODS

Instead of introducing entangled features, most existing poisoning methods rely on generating certain shortcut perturbations that can be more easily learned by the target model than the actual image content (Tao et al., 2021; Segura et al., 2022a; Evtimov et al., 2021; Yu et al., 2022). Figure 5 visualizes the poison representations of different existing methods on an adversarially-trained model. As can be seen, although there exist certain differences between the patterns of different methods (e.g., regarding the relative positions of different classes), all these methods indeed yield well-separable data. This observation also confirms that causing entangled features is a *sufficient condition* to degrade adversarial training.

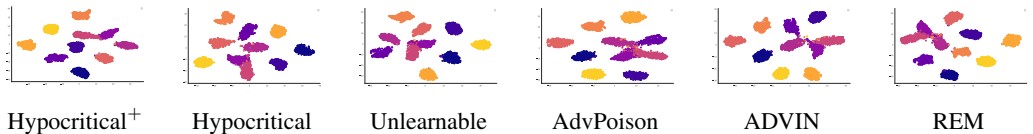

| Hypocritical[+] | Hypocritical | Unlearnable | AdvPoison | ADVIN | REM |

Figure 5: The t-SNE visualizations of the feature representations for existing poisoning methods.

## B    NATURAL AND ADVERSARIAL RISKS

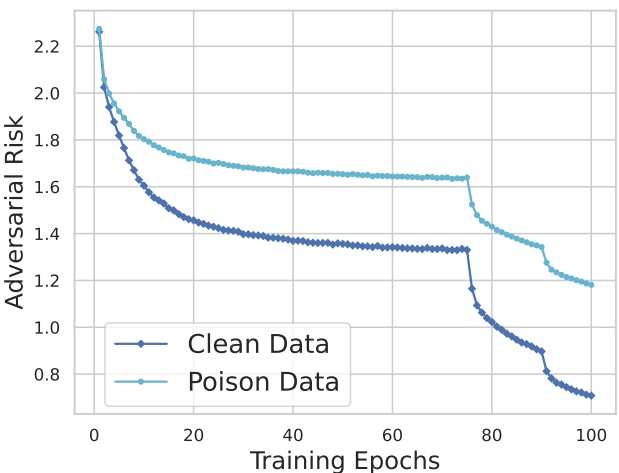

Figure 6: Adversarial risk curves for clean and poison data.

According to Tao et al. (2021), natural risk and adversarial risk are defined as follows.

**Definition 1 (Natural Risk)**

$$\mathcal{R}_{nat}(f, \mathcal{D}) = \mathbb{E}_{(\boldsymbol{x},y)\sim\mathcal{D}}\Big[\ell(f(\boldsymbol{x}), y)\Big]$$

**Definition 2 (Adversarial Risk)**

$$\mathcal{R}_{adv}(f, \mathcal{D}) = \mathbb{E}_{(\boldsymbol{x},y)\sim\mathcal{D}}\Big[\max_{\boldsymbol{x}'\in\mathcal{B}(\boldsymbol{x},\epsilon)} \ell(f(\boldsymbol{x}'), y)\Big]$$

For the whole training process, we calculate the adversarial risk based on Definition 2. The adversarial risks for both clean and poison data are shown in Figure. 6. It validates our claim that adversarial training faces difficulties when optimizing poisoned data because the risk decreases more slowly than the clean case.

## C    DETAILED RESULTS FOR ENTF VS. THE CE LOSS

Table 8: EntF vs. the CE loss using the same robust reference model.

| POISON METHOD \ DATASET | CIFAR-10 | CIFAR-100 | TINYIMAGENET |
|---|---|---|---|
| ADVPOISON-UNTAR | 72.86 | 50.45 | 45.47 |
| ENTF-PUSH | **71.57** | **47.29** | **41.32** |

| (a) Clean (84.88%) | (b) Pull-pair (78.21%) | (c) Pull (72.99%) | (d) Push (71.57%) |

Figure 7: The t-SNE visualizations for EntF-pull-pair vs. our original EntF (Pull and Push).

## D    ADDITIONAL EXPERIMENTAL SETTINGS

All reference and target models are trained for $100$ epochs using SGD optimizer with an initial learning rate of $0.1$ that is decayed by a factor of $0.1$ at the $75$-th and $90$-th training epochs. The optimizer is set with momentum $0.9$ and weight decay $5 \times 10^{-4}$. The inner maximization (i.e., generation of adversarial examples) of the adversarial training is solved by 10-step PGD with a step size of $2/255$.

## E    ADDITIONAL ANALYSIS OF ENTANGLED FEATURES

Our experimental results have demonstrated the effectiveness of EntF in various scenarios. Here we provide additional analysis to better understand the property of entangled features. To this end, we adjust the class selection strategy in EntF-pull to be simply based on pairwise entangled features. Specifically, each pair consists of two classes that have a minimal centroid distance. For optimization, we calculate the centroids of each class pair and then minimize the distance between samples in one class and the centroid of the other class. We denote this attack variant targeting pairwise entangled features as EntF-pull-pair.

We find that EntF-pull-pair can decrease the model accuracy from $84.88\%$ to $78.21\%$, which indicates that introducing pairwise entangled features can already substantially compromise adversarial training. However, there is still a large performance gap between EntF-pull-pair and our original EntF. This can be explained by the fact that EntF-pull exploits more diverse pulling directions based on the sample-class distance, and EntF-push makes samples from multiple classes become overlapped. The visualizations in Figure 7 clearly confirm the above observation that EntF-pull-pair yields entangled features to some extent but still fewer than the original EntF-pull and EntF-push.

## F    ADDITIONAL EXPERIMENTAL RESULTS

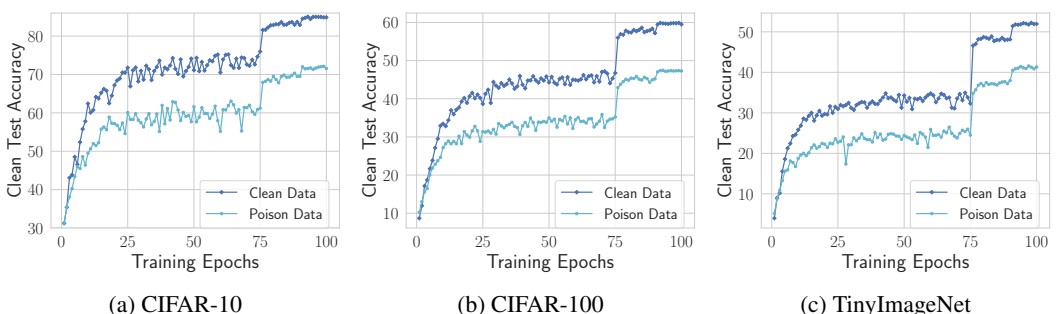

Figure 8: Learning curves of our EntF on different datasets.

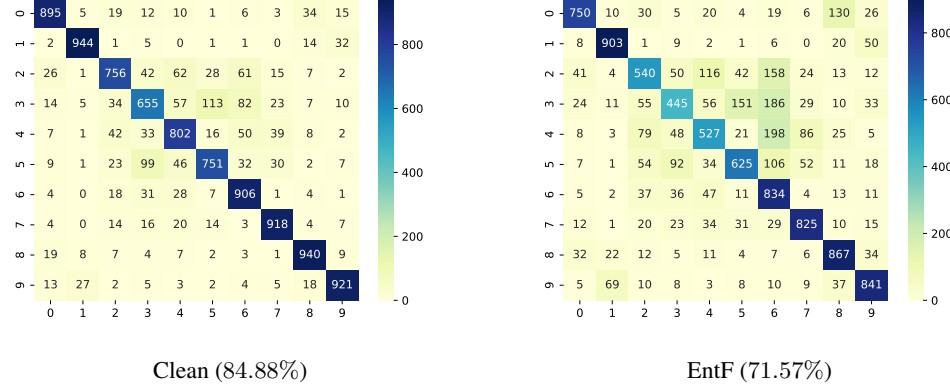

Figure 9: Classification confusion metrics for clean vs. EntF on CIFAR-10. Our EntF causes misclassification from one true class to multiple wrong classes, while previous work causes misclassification from one true class to a dominant wrong class (see Figure 2 in (Fowl et al., 2021b)).

## G  ADDITIONAL PERTURBATION VISUALIZATIONS

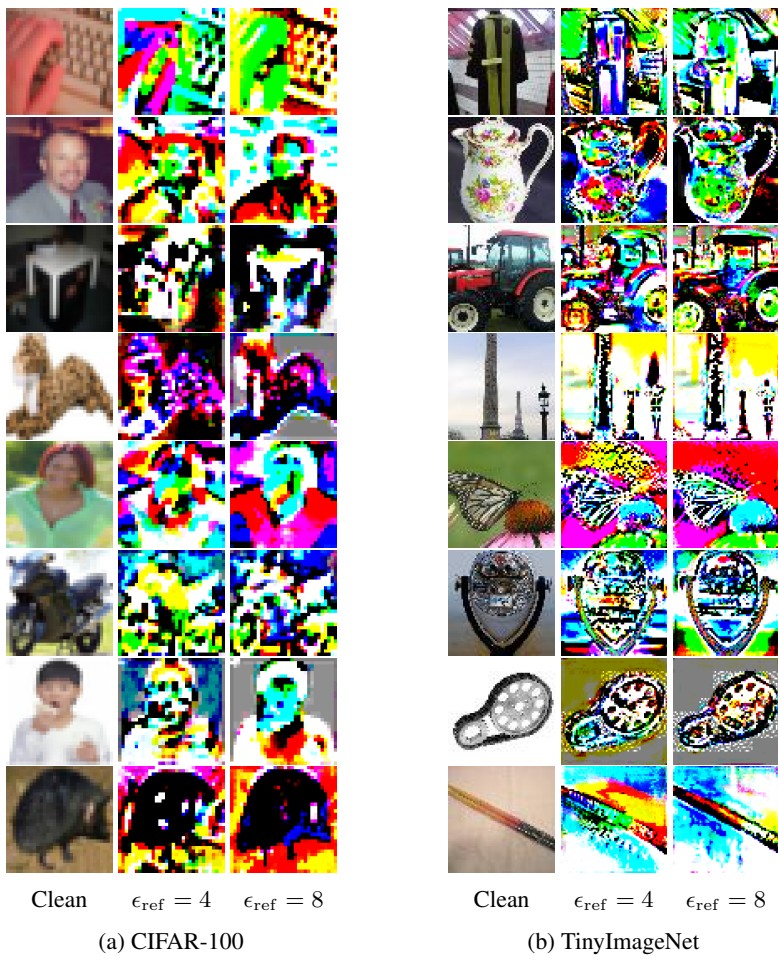

| Clean | $\epsilon_{\mathrm{ref}} = 4$ | $\epsilon_{\mathrm{ref}} = 8$ | | Clean | $\epsilon_{\mathrm{ref}} = 4$ | $\epsilon_{\mathrm{ref}} = 8$ |

(a) CIFAR-100        (b) TinyImageNet

Figure 10: Perturbation (normalized to [0,1]) visualizations for CIFAR-100 and TinyImageNet.

## H  OPTIMAL ACCURACY OF DIFFERENT ATTACKS

Table 9: Attack performance when the defender adjusts AT budget $\epsilon_{\mathrm{adv}}$ to obtain the optimal accuracy.

| POISON METHOD ($\epsilon_{\mathrm{poi}} = 8/255$) | 0/255 | 4/255 | 8/255 | 16/255 | OPTIMAL |
|---|---|---|---|---|---|
| NONE (CLEAN) | 94.59 | 90.31 | 84.88 | 73.78 | 94.59 |
| ADVPOISON | 9.91 | 88.98 | 83.11 | 71.31 | 88.98 |
| REM | 25.59 | **46.57** | 84.21 | 85.76 | 85.76 |
| ADVIN | 77.31 | 90.08 | 86.76 | 72.16 | 90.08 |
| UNLEARNABLE | 25.69 | 90.47 | 84.91 | 79.81 | 90.47 |
| HYPOCRITICAL | 74.06 | 91.18 | 84.96 | 73.33 | 91.18 |
| HYPOCRITICAL[+] | 75.22 | 84.82 | 86.56 | 82.26 | 86.56 |
| ADVPOISON (HYBRID) | **4.16** | 82.74 | 78.65 | 67.17 | 82.74 |
| OURS (HYBRID) | 12.93 | 76.55 | **74.30** | **65.75** | **76.55** |

