# OpenReview forum: "Is Adversarial Training Really a Silver Bullet for Mitigating Data Poisoning?"
_ICLR.cc/2023/Conference — ICLR 2023 notable top 25%_

### Official Review · Reviewer_DCRE · 2022-10-24

**Confidence:** 4
**Clarity, Quality, Novelty And Reproducibility:** This paper is relatively well written…
**Correctness:** 3
**Technical Novelty And Significance:** 3
**Empirical Novelty And Significance:** 3
**Recommendation:** 6

**Strength And Weaknesses:**

Strengths:
* The method is novel and obtains state-of-the-art levels of poisoning success on CIFAR-10 adversarially trained models. The authors' attack is interesting: given a learning algorithm, it exploits the model class' idiosyncrasies to poison newly trained models.

Weaknesses (mostly in writing and unnecessary claims):
* Claims about the mechanism of INF (and of other attacks). Any claims about the claims need to be properly defined or removed. For example, from the introduction "Different from our attack strategy, existing methods commonly inject perturbations as shortcuts, which ensures that the model wrongly learns the shortcuts rather than the actual image content, leading to low test accuracy (Segura et al., 2022a; Evtimov et al., 2021; Yu et al., 2022)" - there is no definition of what a shortcut or indiscriminate feature here is. The authors should describe concretely in what sense the given attack is different from previous work. These poorly defined claims around shortcuts vs indescriminate features are throughout the paper and should be rectified (or removed) before the work is accepted.
* Poor contextualization: Is adversarial training currently considered the best possible defense in the given $\ell_p$-based threat model? It would be good to provide this kind of context.

**Summary Of The Paper:**

In this work the authors attack in models in the following adversarial threat model:
* Problem considered: CIFAR-10 image classification task
* Attacker threat model: can modify each training image by up to $\epsilon=8/255$ in $\ell_\inf$ norm
* Attacker goal: minimize clean test accuracy on models trained on this dataset

The authors develop a new algorithm, INF, for this kind of data poisoning task, specifically aimed at the defense of performing adversarial training at train time. The authors find that they can decrease accuracy by 13.31% (from 84.88% under no poisoning to 71.57% under INF).

**Summary Of The Review:**

The paper presents a new method that yields state-of-the-art poisoning on adversarially trained models, which were previously seen as an effective defense against $\ell_p$-based data poisoning methods. The paper has some issues around claims and contextualization of work,.

---

> ### Author Response · Authors · 2022-11-08
> **Response to Reviewer DCRE**
>
> **Q. A strong baseline based on class-wise random perturbations.**
>
> **A.** We were surprised to see the very strong result reported by the reviewer since existing work has proved the **opposite** result, i.e., class-wise random perturbations fail in poisoning AT models (See P5: Universal random perturbations in [1]). After checking the code provided by the reviewer, we find the result is actually wrong, due to one obvious **implementation error**. Specifically, in line 27,
> ```python
> data = data.astype(int) * BUDGET
> ```
> , the BUDGET was mistakenly multiplied by the data. This operation dramatically amplifies the poison budget. For example, if an original pixel value is 30, the value after the multiplication becomes 30*8 = 240, which is clearly out of the allowed pixel range, [30-8, 30+8]. After fixing this error by replacing line 27 to,
> ```python
> data = data.astype(int)
> poisons = poisons.astype(int) * BUDGET
> ```
> , the result turns from 20.83% to 84.06%, suggesting the negligible impact of class-wise random perturbations on the original model accuracy, i.e., 84.88%.
>
> More generally, class-wise poisons are known to be practically unfavorable because they can be easily recovered by taking the average image of a class [2]. This is also the main reason why the state-of-the-art attack methods, as compared in our paper, are all **sample-wise**. We also tried generating random perturbations in a more meaningful, sample-wise manner, and the result is 84.26\%, which also does not work.
>
> We have added this random perturbation baseline to Table 1 in the revised paper.
>
> [1] Better Safe Than Sorry: Preventing Delusive Adversaries with Adversarial Training. NeurIPS 2021
>
> [2] Autoregressive Perturbations for Data Poisoning. NeurIPS 2022.

---

> > ### Comment · Reviewer_DCRE · 2022-12-03
> > **Re: response**
> >
> > Thank you for catching this issue, you are definitely right!

---

> > > ### Author Response · Authors · 2022-12-06
> > > **Further response to Reviewer DCRE**
> > >
> > > Thank you for your response, and we are delighted that we have addressed your concern about the baseline attack.
> > >
> > > > There is no definition of what a shortcut or indiscriminate feature here is.
> > >
> > > "Shortcut" commonly refers to spurious correlations learned between model inputs, e.g., image features, and outputs, e.g., image labels, in shortcut learning [1]. This term has been adopted by recent work on poisoning [2]. That is to say, existing poisoning methods focus on causing spurious correlations between image perturbations and class labels.
> > >
> > > "Indiscriminative features" means that features from two (or multiple) classes become indiscriminative (or overlapped/entangled), which is operated in the feature space and fundamentally different from causing spurious correlations.
> > >
> > > In the final version, we will explicitly define these two terms and describe their differences.
> > >
> > > [1] Shortcut learning in deep neural networks. Nature Machine Intelligence 2020.
> > >
> > > [2] Availability Attacks Create Shortcuts. KDD 2022
> > >
> > > > It would be good to provide the context about whether adversarial training is currently considered the best possible defense in the given threat model.
> > >
> > > Thanks for your suggestion. Although the "Silver Bullet" in our title has somehow reflected this point, we found that it is indeed not explicitly mentioned in our paper. Yes, adversarial training is the best possible defense in the given threat model and its effectiveness (against existing attacks) has been supported both theoretically and empirically.
> > >
> > > In the final version, we will make this point clear, especially in the Abstract, Introduction, and Methodology sections.

---

> ### Author Response · Authors · 2022-11-15
> **Awaiting reviewer interactions**
>
> Dear Reviewer DCRE,
>
> Thank you very much for your comments. We have been eagerly awaiting your interaction about our response to your initial concern. Please also let us know if you have any further questions, and we are very happy to follow up!

---

### Official Review · Reviewer_ffZU · 2022-10-24

**Confidence:** 4
**Correctness:** 4
**Technical Novelty And Significance:** 3
**Empirical Novelty And Significance:** 3
**Recommendation:** 8

**Clarity, Quality, Novelty And Reproducibility:**

- The paper is well-organized, and the proposed attack method is clearly introduced.
- The proposed attack method is novel and sound.
- The authors conduct various quantitative evaluations and qualitative visualizations, which provide insights for the proposed method.

**Strength And Weaknesses:**

Strengths:
- The setting of $\epsilon_{adv}\ge \epsilon_{poi}$ considered in this paper is indeed more reasonable than the setting of $\epsilon_{poi}\ge 2\epsilon_{adv}$.
- The paper points out an implicit assumption behind the defense ability of adversarial training against clean-label availability poisoning attacks: adversarial training is capable of minimizing the adversarial risk on the poisoned data distribution. When this assumption is broken by the proposed attack, adversarial training would fail.
- Experimental results are strong. As far as I know, no previous attack can degrade the test accuracy of adversarially trained models by more than 2% on CIFAR-10. In contrast, the proposed attack can degrade the test accuracy by more than 10%. This is impressive.


Weaknesses:

- The authors claim that the proposed attack method makes some features become not useful even for adversarial training, and thus the assumption behind the defense ability of adversarial training is broken. This means that when the proposed method induces indiscriminative features in the training data, adversarial training becomes not capable of minimizing the adversarial risk on the poisoned data. Are there any experimental results that validate that adversarial training faces difficulties in minimizing the adversarial risk of such poisoned data?

- It is interesting to see that in Table 3, when the poison budget $\epsilon_{poi}=8/255$, adversarial training with $\epsilon_{adv}=4/255$ performs better than adversarial training with $\epsilon_{adv}=8/255$. Could you provide any insight about this?

- The reference model adopted in this paper is adversarially trained. Figure 4 shows that this performs better than adopting naturally trained models. The effectiveness of the previous attack methods may also be improved by adopting an adversarially trained model as the reference model. Have the previous works tried this? Do you have any results to share?


**Summary Of The Paper:**

This paper challenges the defense ability of adversarial training against clean-label availability poisoning attacks. It was believed that these attacks can hardly harm adversarially trained models. However, the proposed attack method in the paper substantially degrades the test accuracy of adversarially trained models from 84.88% to 71.57% on CIFAR-10, which is seven times better than existing methods, resulting in a new SOTA.


**Summary Of The Review:**

This paper establishes a new SOTA in degrading the test accuracy of adversarial training under clean-label availability attacks. Overall, the proposed method is adequately evaluated, and the results are very impressive. Thus, the paper successfully challenges the consensus that clean-label availability attacks can hardly harm adversarial training.

---

> ### Author Response · Authors · 2022-11-08
> **Response to Reviewer ffZU**
>
>
> **Q1. Experimental results for validating that adversarial training faces difficulties in minimizing the adversarial risk.**
>
> **A1.** We compare the adversarial risk of a poisoned AT model to that of a clean AT model.
> Here the adversarial risk is calculated following the formula from [1]:
>
> $\mathcal{R}\_\text{adv}(f,\mathcal{D})=\mathbb{E}\_{(\boldsymbol{x},y)\sim\mathcal{D}}[\max_{\boldsymbol{x'}\in\mathcal{B}(\boldsymbol{x},\epsilon)}\ell(f(\boldsymbol{x'}),y)]$.
>
> We find that compared to the clean training, our poisoned method yields a much higher adversarial risk (1.182 vs. 0.708). This result validates our claim that our poisoning method makes it harder for the AT model to minimize the adversarial risk.
>
> We have shown the detailed results in Figure 6 of the revised paper.
>
> **Q2. Insight about the finding that AT with $\varepsilon_{adv}=4/255$ performs better than AT with $\varepsilon_{adv}=8/255$ against the proposed attack when $\varepsilon_{poi}=8/255$.**
>
> **A2.** This is indeed an interesting finding, which can be explained as follows. On the one hand, enlarging the AT budget can increase clean accuracy due to the increased robustness to poisons. On the other hand, enlarging the AT budget inevitably leads to an accuracy drop compared to standard training. These two **conflicting** factors indicate that the model accuracy and AT budget do not have a clear correlation.
>
> We have highlighted this point in the revised version and left more detailed explorations for future work.
>
> **Q3.  Using AT reference models for baseline methods.**
>
> **A3.** In Table1, Hypocritical+ is a variant of Hypocritical with an AT reference model, REM is the AT variant of Unlearnable, and ADVIN also uses an AT reference model. We additionally try an AT variant of AdvPoison, which only decreases the model accuracy to 83.69%. In addition, under the “Important note on the cross-entropy loss” of Section 3, we have also discussed the superiority of our loss function over the commonly-used cross-entropy loss when the same AT reference model is used. All these results support that adopting an AT reference model is not the only factor that leads to the success of our method.
>
> We have added the reference model information of different methods to the caption of Table 1 in the revised paper.
>
> [1] Better Safe Than Sorry: Preventing Delusive Adversaries with Adversarial Training. NeurIPS 2021

---

> > ### Comment · Reviewer_ffZU · 2022-12-02
> > **Thanks**
> >
> > I appreciate the authors for the response. My concerns are well addressed.
> >
> > After reading other reviewers' comments, I thought it would be helpful if the authors could report the results of the compared attacks under lower and higher AT budgets. For example, I would like to know whether the compared attacks (such as REM, AdvPoison, and Hyporitical) perform worse when $\epsilon_{adv}=\epsilon_{poi}/2=4/255$.

---

> > > ### Author Response · Authors · 2022-12-02
> > > **Further response to Reviewer ffZU**
> > >
> > > Thanks for your feedback and we are glad that our previous response has well addressed your concerns.
> > >
> > > Based on our understanding, you are curious about other attacks’ performance when the defender is able to adjust their AT budget  $\epsilon_\mathrm{adv}$ to find an *optimal* accuracy against poisoning under a specific $\epsilon_\mathrm{poi}$.
> > >
> > > We conduct additional experiments and report the results in the following table. In particular, the last column reports the optimal accuracy the practical defender can achieve.
> > >
> > > | Method ($\epsilon_\mathrm{poi}=8/255$) $\backslash$ $\epsilon_\mathrm{adv}$ | $0/255$                | $4/255$                | $8/255$                | $16/255$               | Optimal Accuracy       |
> > > |:------------------------------------:|:------------------------:|:------------------------:|:------------------------:|:------------------------:|:------------------------:|
> > > | None (Clean)                       | $94.59$                | $90.31$                | $84.88$                | $73.78$                | $94.59$                |
> > > | AdvPoison                          | $\textbf{9.91}$  | $88.98$ | $83.11$ | $71.31$ | $88.98$ |
> > > | REM                          | $25.59$ | $\textbf{46.57}$ | $84.21$ | $85.76$ | $85.76$ |
> > > | ADVIN                         | $77.31$   | $90.08$ | $86.76$ | $72.16$ | $90.08$ |
> > > | Unlearnable                         | $25.69$   | $90.47$ | $84.91$ | $79.81$ | $90.47$ |
> > > | Hypocritical                       | $74.06$   | $91.18$ | $84.96$ | $73.33$ | $91.18$ |
> > > | Hypocritical+                     | $75.22$   | $84.82$ | $86.56$ | $82.26$ | $86.56$ |
> > > | **Ours (Hybrid)**                           | $11.89$ | $76.55$ | $\textbf{74.30}$ | $\textbf{65.75}$ | $\textbf{76.55}$ |
> > >
> > > As can be seen, our (hybrid) attack fools the defender to achieve the lowest optimal accuracy, compared to other attacks. Note that this hybrid attack is specifically designed to be globally effective against an adjustable defender, and more details can be found in our paper and the response to Reviewer VPK4. We will add these results to the final version of our paper.
> > >
> > > It is worth noting that the most signification contribution is that we have challenged the consensus that adversarial training can be used as the principal defense against poisoning attacks, thereby removing the false sense of security.
> > >
> > > The latest version of our paper has been divided into two parts, with the first part sticking to the contribution of challenging the consensus and the second part aimed to provide a globally effective attack that can degrade the clean model accuracy when the defender is able to adjust the adversarial training budgets.

---

> > > > ### Comment · Reviewer_ffZU · 2022-12-06
> > > > **Thanks**
> > > >
> > > > Thanks for the additional experiments. The results show that the proposed method performs best even when the defender adjusts the defense budget. I thought such results make the evaluation of the proposed attack more thorough. Thus, the authors are encouraged to include these in the revision.
> > > >
> > > > I would like to support this work, not only because it achieves SOTA performance in a reasonable setting, but also because this setting is more reasonable than the setting in recent works.
> > > >
> > > > There were recent works whose main claim is that their attacks break through the defense of adversarial training in the setting of $\epsilon_{poi}>2\epsilon_{adv}$. In such a setting, it is somewhat trivial to claim that adversarial training is bypassed. Since adversarial training is only guaranteed to defend against the poisons with $\epsilon_{poi} \le \epsilon_{adv}$, it is not surprising that it fails when $\epsilon_{poi}>2\epsilon_{adv}$. Instead, this work surprises me in that the proposed attack substantially degrades the test accuracy of adversarial training in the setting of $\epsilon_{poi}=\epsilon_{adv}$. This successfully challenges the consensus that clean-label availability attacks can hardly harm adversarial training.

---

> > > > > ### Author Response · Authors · 2022-12-06
> > > > > **Thank you**
> > > > >
> > > > > Thank you for your efforts in reviewing our paper. We are happy that our response has addressed your concerns and grateful for your valuable suggestions, which definitely make our work better. We will update our paper as promised in our response.

---

### Official Review · Reviewer_VP8n · 2022-10-25

**Confidence:** 4
**Correctness:** 4
**Technical Novelty And Significance:** 4
**Empirical Novelty And Significance:** 3
**Recommendation:** 10

**Clarity, Quality, Novelty And Reproducibility:**

The paper is very clear and well written.  It seems reducible, though I have not looked at the code.

**Strength And Weaknesses:**

**Strengths**

- The paper is very clear and easy to read.
- The approach is intuitive and effective.
- The problem is relevant.
- Visualizations are clear and informative.

**Weaknesses**

The weaknesses are few. It's impact will not be world-changing, but it is a professionally written paper.  The description of Theorem 1 is awkward, as it uses notation and terms from another paper that is not introduced in this paper (for example, "risk" and $\mathcal{R}$).  This should be easily remedied.

**Summary Of The Paper:**

The paper introduces a new approach, indiscriminative features (INF), to the problem of poisoning a data set so as to make predictive models learned from the data less successful, even if those models are trained using adversarial training.  Typically, adversarial training has been a successful defense against indiscriminative data poisoning.  To overcome this, the paper introduces INF-push and INF-pull.  INF-push permutes the data points so their points' features in a learned model are maximally moved further from the centroid of their class's centroid, within a permutation budget.  In INF-pull, data points are permuted so their points' features are moved closer to the nearest centroid of an incorrect class.

The paper demonstrates these poisons effectively evade a defense of adversarial training, as well as several other reasonable defenses.

**Summary Of The Review:**

A very good and interesting paper, which moves the ball forward on indiscriminate poisoning.

---

> ### Author Response · Authors · 2022-11-08
> **Response to Reviewer VP8n**
>
> Thanks for the reviewer’s compliment on our work. We have added definitions of those notations in the revised paper.

---

### Official Review · Reviewer_VPK4 · 2022-10-28

**Confidence:** 5
**Clarity, Quality, Novelty And Reproducibility:** The writing is clear, the method is n…
**Correctness:** 3
**Technical Novelty And Significance:** 3
**Empirical Novelty And Significance:** 4
**Recommendation:** 6

**Strength And Weaknesses:**

### Pros:
1) The attack is intuitive, and the results appear promising.
2) The paper is clear and well written.
3) Experimentation is fairly thorough, and a large number of related works are compared to.

### Cons/Random things to address:
1) I strongly encourage the authors to re-name their method and rephrase its positioning. The authors claim that their poisoning method is successful because it perturbs "indiscriminative features". However, in my opinion, there is little basis for this claim. It is also an unnecessary claim as the paper's results ostensibly justify the proposed method. In fact, I would argue that the poisoning technique does in fact perturb discriminative features - especially in the INF pull case as the perturbations by construction center the features of the poisoned data around the class mean - a descriminative quantity... Observing the visualizations of your perturbations would seem to confirm this. To their credit, the authors do try to justify this claim (briefly) in section 6, but TSNE visualizations are not convincing evidence of this claim. It's also not clear from what models the plots in Figure 5 were generated. Are they the features according to the reference model or the victim model after poisoning?
2) Table 1 presents very encouraging results for your method. However, from Table 3, it appears that your attack success is reduced by using a smaller adversarial training $\varepsilon$-budget. To present a fair comparison, Table 1 should instead show the *maximum* adversarial training accuracy a practitioner could achieve against each method - a practitioner would not use a higher than necessary adversarial training budget if a smaller one produces better results.
3) If I understand correctly, there is another slightly unfair aspect to the comparisons to previous works. Your method crafts the perturbations using an adversarially trained reference model, and you show the superiority of your method compared to other methods in the face of an adversarial training defense. Did you adapt the other attacks and use an adversarially trained model for those results? If not, Table 1 doesn't present an apples-to-apples comparison. As you state "More specifically, modifying the robust (semantic) features is the key to poisoning the AT models". Maybe this is key for other methods as well.
4) While I understand your point that defenders might not have access to a clean-trained model, or a clean AT model, I think this is a bit of a stretch. What if a defender could use a general purpose clean-trained feature extractor (widely available) to defend against your attack?
5) Potentially the biggest issue I have is Figure 4a. If I'm reading this correctly, if the defender simply trained  in a standard fashion (not AT), the results in Table 1 would jump well into the 80's? Why would the defender choose to adversarially train then?

Overall, I think the paper is interesting, and I'm willing to raise my score if my concerns are sufficiently addressed.





**Summary Of The Paper:**

The authors propose a novel technique to poison training data by crafting perturbations to either repel or attract training data to the corresponding class' mean feature vector. The authors demonstrate that this poisoning technique is superior to known availability attacks when defended against with adversarial training.

**Summary Of The Review:**

I find the work interesting and well motivated, but I have serious questions about the setting for comparisons. I am willing to raise my score if these questions are addressed.

---

> ### Author Response · Authors · 2022-11-08
> **Response to Reviewer VPK4 (1/2)**
>
> **Q1.The name of our method and t-SNE plots in Figure 5.**
>
> **A1.** There seems to be a misunderstanding. We have not claimed that our method perturbs “indiscriminative features” but instead, it **induces** “indiscriminative features”. Our claim is consistent with the reviewer's comment that discriminative features are perturbed (to become indiscriminative). All t-SNE plots in our paper were generated using the same clean AT reference model with $\epsilon_\mathrm{ref}=4$.
>
> We have added this information to the caption of Figure 1 in the revised paper.
>
> **Q2. Table 1 should report the optimal result for each attack method. In particular, for the proposed attack, the result for $\epsilon_\mathrm{adv}=4$ should be reported.**
>
> **A2.** We argue that Table 1 should not report the optimal AT setting for each attack but the globally optimal setting, i.e. $\epsilon_\mathrm{adv}=8$. A strong defender is supposed to achieve **global** robustness against all possible attacks under a specific threat model, here $\epsilon_\mathrm{poi}=8$ for Table 1. It is also not feasible for a practitioner to adapt their defense to every single attack. According to [1], the robustness guarantee of AT against poisoning attacks holds only when $\epsilon_\mathrm{adv}\geq \epsilon_\mathrm{poi}$. As a result, it is not intuitive for a practitioner to set $\epsilon_\mathrm{adv}< \epsilon_\mathrm{poi}$. More specifically, as we have stated in our paper, REM and ADVIN are highly effective when $\epsilon_\mathrm{poi}=2 \epsilon_\mathrm{adv}$.
>
> Regardless of the above practical consideration, it is indeed an interesting finding that the AT model with $\epsilon_\mathrm{adv}=4$ performs better than AT with $\epsilon_\mathrm{adv}=8$ against our attack when $\epsilon_\mathrm{poi}=8$. We explain this finding as follows. On the one hand, enlarging the AT budget can increase clean accuracy due to the increased robustness to poisons. On the other hand, enlarging the AT budget inevitably leads to an accuracy drop compared to standard training. These two **conflicting** factors indicate that the model accuracy and AT budget do not have a clear correlation.
>
> We have highlighted this point in the revised version and left more detailed explorations for future work.
>
> [1] Better Safe Than Sorry: Preventing Delusive Adversaries with Adversarial Training. NeurIPS 2021
>
> **Q3. Using AT reference models for baseline methods.**
>
> **A3.** In Table1, Hypocritical+ is a variant of Hypocritical with an AT reference model, REM is the AT variant of Unlearnable, and ADVIN also uses an AT reference model. We additionally try an AT variant of AdvPoison, which only decreases the model accuracy to 83.69%. In addition, under the “Important note on the cross-entropy loss” of Section 3, we have also discussed the superiority of our loss function over the commonly-used cross-entropy loss when the same AT reference model is used. All these results support that adopting an AT reference model is not the only factor that leads to the success of our method.
>
> We have added the reference model information of different methods to the caption of Table 1 in the revised paper.
>
> **Q4. Adaptive defense with a general-purpose clean-trained feature extractor.**
>
> **A4.** Following the reviewer’s suggestion, we test general-purpose clean-trained feature extractors and find that they cannot help mitigate our poisons.
> Specifically, we test whether feature extractors that are trained on clean CIFAR-100 data can help mitigate CIFAR-10 poisons. We consider both ST and AT training and make sure the feature extractors share the same architecture as the CIFAR-10 reference model used in our attack. These two feature extractors are used to pull the CIFAR-10 poisoned samples back towards their original class centroids.
>
> We find that the ST feature extractor changes the model accuracy from 71.57\% to 71.53\%. This poor result is expected based on our finding in Section 5 that poisoning ST and AT models require modifying different types of features. In other words, an ST feature extractor cannot help mitigate AT poisons. For the AT feature extractor, although the same type of features is used, the result is still very low (72.97%).
>
> We have added the above results in the revised paper.

---

> > ### Author Response · Authors · 2022-11-08
> > **Response to Reviewer VPK4 (2/2)**
> >
> > **Q5. Figure 4: Since an ST defender is effective against AT poisons, why bother training an AT defender?**
> >
> > **A5.** A strong defense aims to achieve **global** robustness to all possible attacks under a specific threat model but not to only one specific attack.
> >
> > Although the ST defender can mitigate AT poisons, including ours, to some extent, it cannot serve as a globally strong defense because it is highly vulnerable to ST poisons. Figure 4 (a) clearly shows that our ST poisons can reduce the ST model’s accuracy to a random guess level. Moreover, existing attacks are also very good at poisoning ST models. On the other hand, the AT defender was previously thought to be globally strong, but our work challenges this consensus.
> >
> > Our Section 5 aims to uncover the above trade-off between using robust vs. non-robust features to poison AT vs. ST models. Inspired by this comment, we would like to shed more light on this trade-off by trying a poisoning solution that could be effective against both the ST and AT models. Specifically, for each image, we simply take an average of the non-robust and robust perturbations, which ensures that the poison budget remains unchanged. This leads to an accuracy of 22.86% for poisoning an ST model and 77.19% for poisoning an AT model. These results indicate that it’s possible to poison ST and AT simultaneously. We leave more detailed explorations to find the optimal combination strategy for future work.
> >
> > We have added a discussion about the above combination method in the revised paper.

---

> > > ### Comment · Reviewer_VPK4 · 2022-11-12
> > > **Response (2/2)**
> > >
> > > > A5. A strong defense aims to achieve global robustness to all possible attacks under a specific threat model but not to only one specific attack.
> > >
> > > Again, I disagree with this setting as a benchmark. A much better setting in which to measure success would be for the defender to take $max_{\varepsilon_{adv}}$ of the model accuracy. The only world in which your original proposed setting makes sense is one in which the defender knows their data has been poisoned, but doesn't know by which method, and has zero amount of clean data on which to test. If any of these three conditions are not met, then the defender easily bypasses your proposed attack.
> > >
> > > As an analogy, if a work proposes a new method for adversarial attacks which always fools adversarially-trained models, but doesn't work when the model isn't adversarially trained, this would not be considered a strong attack.
> > >
> > > I would say in fact that it is the attacks have to be *globally robust* to at least naive defenses.
> > >
> > > &nbsp;
> > >
> > > > the AT defender was previously thought to be globally strong, but our work challenges this consensus.
> > >
> > > I agree more with this claim, and I think this is the stronger framing of your work, instead of as an improved attack (for the reasons listed in the above response).
> > >
> > > &nbsp;
> > >
> > > > which ensures that the poison budget remains unchanged. This leads to an accuracy of 22.86% for poisoning an ST model and 77.19% for poisoning an AT model
> > >
> > > This is good to see, and somewhat mitigates my concerns with the initial plot. I definitely think this should be highlighted in the paper. Could you produce a table which shoes the success of these poisons under different $\varepsilon_{adv}$ and $\varepsilon_{poison}$ choices?

---

> > > > ### Author Response · Authors · 2022-11-15
> > > > **Further response to Reviewer VPK4**
> > > >
> > > > > The "indiscriminative" claim should be rephrased regardless of the clear technical contributions.
> > > >
> > > > We appreciate your further clarification about your understanding of our "indiscriminative" claim. Based on your example, now we get where your concern comes from. Actually, "indiscriminative features" in our paper means that features from two (or multiple) classes become indiscriminative (or overlapped/entangled) rather than the discriminative ability of one single feature. However, we have not explicitly mentioned this context every time we say "indiscriminative features". This is indeed a problem. To address this problem, we have planned to use “entangled features”, which naturally imply the relationship of multiple classes. Please feel free to give other suggestions or comments on the rephrasing.
> > > >
> > > > By the way, technically, even for INF-Pull, the perturbations might not induce the same feature for every input in a given class because every input is pulled towards the centroid of **its own** nearest class.
> > > >
> > > > > Static to the original AT claim for Table 1 rather than using the claim of globally robust defense.
> > > >
> > > > Thanks for your suggestion and detailed explanations. We agree with you on this point. In the revised paper, we have also made several modifications to Section 4 to make sure our claims focus on challenging the consensus about the effectiveness of AT.
> > > >
> > > > > Results of the hybrid attack should be highlighted, with a table showing the success of poisons under different $\epsilon_\mathrm{adv}$ vs. $\epsilon_\mathrm{poi}$.
> > > >
> > > > We have added new content in Section 5 to discuss two possible hybrid attacks in detail and highlighted this contribution through the whole paper. Table 7 in Section 5 clearly shows that both our two hybrid attacks substantially compromise a defender that can vary their AT budget $\epsilon_\mathrm{adv}$ from 0 (i.e., ST) to 16. More specifically, the optimization-based hybrid attack outperforms the average-based one by a large margin. For ease of reference, we list Table 7 in the following. Due to the page limit, we have moved the original Section 6 to the appendix.
> > > >
> > > > **Table 7** Evaluating two hybrid attacks (average/optimization) under different $\epsilon_\mathrm{poi}$ vs. $\epsilon_\mathrm{adv}$.
> > > > | $\epsilon_\mathrm{poi}$ $\backslash$ $\epsilon_\mathrm{adv}$ | $0/255$                | $4/255$                | $8/255$                | $16/255$               | Average                |
> > > > |------------------------------------|------------------------|------------------------|------------------------|------------------------|------------------------|
> > > > | None (Clean)                       | $94.59$                | $90.31$                | $84.88$                | $73.78$                | $85.89$                |
> > > > | $4/255$                            | $51.98/	\textbf{27.66}$  | $86.87/\textbf{85.85}$ | $81.24/\textbf{80.90}$ | $71.13/\textbf{70.76}$ | $72.81/\textbf{66.29}$ |
> > > > | $8/255$                            | $22.86/\textbf{12.93}$ | $82.30/\textbf{78.89}$ | $77.19/\textbf{75.06}$ | $\textbf{66.70}/66.76$ | $62.26/\textbf{58.41}$ |
> > > > | $16/255$                           | $5.34/\textbf{4.80}$   | $75.32/\textbf{63.86}$ | $69.41/\textbf{66.25}$ | $\textbf{59.40}/60.73$ | $52.37/\textbf{48.91}$ |

---

> > > > ### Author Response · Authors · 2022-12-02
> > > > **Update about the global attack**
> > > >
> > > > > Additional results about the global attack effectiveness (related to your initial concern 2).
> > > >
> > > > As requested by Reviewer ffZU, we additionally evaluate other attacks against a defender that can adjust their AT budgets. As can be seen in the following table, our (hybrid) attack fools the defender to achieve the lowest optimal accuracy.
> > > >
> > > > | Method ($\epsilon_\mathrm{poi}=8/255$) $\backslash$ $\epsilon_\mathrm{adv}$ | $0/255$                | $4/255$                | $8/255$                | $16/255$               | Optimal Accuracy       |
> > > > |:------------------------------------:|:------------------------:|:------------------------:|:------------------------:|:------------------------:|:------------------------:|
> > > > | None (Clean)                       | $94.59$                | $90.31$                | $84.88$                | $73.78$                | $94.59$                |
> > > > | AdvPoison                          | $\textbf{9.91}$  | $88.98$ | $83.11$ | $71.31$ | $88.98$ |
> > > > | REM                          | $25.59$ | $\textbf{46.57}$ | $84.21$ | $85.76$ | $85.76$ |
> > > > | ADVIN                         | $77.31$   | $90.08$ | $86.76$ | $72.16$ | $90.08$ |
> > > > | Unlearnable                         | $25.69$   | $90.47$ | $84.91$ | $79.81$ | $90.47$ |
> > > > | Hypocritical                       | $74.06$   | $91.18$ | $84.96$ | $73.33$ | $91.18$ |
> > > > | Hypocritical+                     | $75.22$   | $84.82$ | $86.56$ | $82.26$ | $86.56$ |
> > > > | **Ours (Hybrid)**                           | $11.89$ | $76.55$ | $\textbf{74.30}$ | $\textbf{65.75}$ | $\textbf{76.55}$ |
> > > >
> > > >
> > > > Note that here the hybrid attack uses three reference models (with $\epsilon_\mathrm{ref}=0$, $\epsilon_\mathrm{ref}=2$, $\epsilon_\mathrm{ref}=4$), and it further boosts the performance of the hybrid attack in the above Table 7, which uses two references model.

---

> > ### Comment · Reviewer_VPK4 · 2022-11-12
> > **Response**
> >
> > Thanks for the detailed response.
> >
> > > A1. There seems to be a misunderstanding.
> >
> > There isn't any misunderstanding - I just didn't phrase my concern well, so I apologize for that. We actually are talking about the same thing. That being said, I do not agree that your perturbations - at least the INF-Pull method - induces indiscriminate features. Take the limit as the perturbation budget ($\varepsilon$) grows. Then almost by construction your perturbations induce the *same* feature for every input in a given class, which is clearly discriminative. A t-sne plot is not strong justification for your claim in my eyes, especially when it goes against the above intuition.
> >
> > However, I would like to make it clear that I think there is merit to the work as a new objective regardless of this "indiscriminative" claim - I just don't buy your framing of the work, and strongly urge you to rephrase.
> >
> > &nbsp;
> >
> > > A2. We argue that Table 1 should not report the optimal AT setting for each attack but the globally optimal setting,
> >
> > What does "globally optimal" mean? Does this mean the average robustness taken over the listed attacks? This seems like a strange "benchmark". How does the defender know that the poisoner used $\varepsilon=8$? It seems to me the most realistic scenario for a defender is that they have a small set of clean samples on which to validate training, and if the defender suspects their training data has been poisoned, they would test several levels of $\varepsilon$ and choose the optimal one as validated on their small clean dataset.
> >
> > To this end, it seems like then, Table 1 operates in a gray-box type setting wherein the defender knows their data has been poisoned, but don't know which method. If you keep Table 1 in this framing, you should also include a Table for white-box settings where the defender knows which method was used to poison their data - in which case they could easily choose a smaller $\varepsilon$ value with which to defend if this is known to be superior for mitigating your poisons.
> >
> > &nbsp;
> >
> > > the robustness guarantee of AT against poisoning attacks holds only when $\varepsilon_{adv} \geq \varepsilon_{poison}$. As a result, it is not intuitive for a practitioner to set $\varepsilon_{adv} < \varepsilon_{poison}$
> >
> > Again, how does the defender know what $\varepsilon$ budget the poisoner used? If this is a white box Table, then it's reasonable to assume the defender knows which attack is used.
> >
> > &nbsp;
> >
> > >A3
> >
> > Thanks for the clarification!

---

> ### Author Response · Authors · 2022-11-18
> **Satisfied with our response?**
>
> Dear Reviewer VPK4,
>
> We have tried to address your comments raised in the original and the follow-up reviews, and all corresponding revisions are marked in blue in our revised paper.
>
> In particular, we have planned to address your concern about “indiscriminative” by using “entangled features” to name our method. We are not allowed to update the paper since tomorrow (Nov, 19). However, if you have other suggestions, we promise we will incorporate them in the final version.

---

> > ### Comment · Reviewer_VPK4 · 2022-12-09
> > **Response**
> >
> > Apologies for the longish delay. Thank you for your updates and your engagement during the rebuttal. I am happy with the hybrid attack you introduce. I have therefore raised my score. However, I would strongly encourage the authors to include adaptations of other attacks to this "hybrid" setting and report results, and also report the *strongest* defense against each attack in Table 1.

---

> > > ### Author Response · Authors · 2022-12-11
> > > **Thank you for your response**
> > >
> > > Thank you for your efforts in the whole review process, especially the interactions in the response phase. We are happy to see our response has addressed your concerns, and we appreciate that you have raised the score.
> > >
> > > > I would strongly encourage the authors to include adaptations of other attacks to this "hybrid" setting and report results.
> > >
> > > We apply our hybrid technique to AdvPoison (the best existing attacks according to our Table 1), and report the results in the following table.
> > >
> > > | Method ($\epsilon_\mathrm{poi}=8/255$) $\backslash$ $\epsilon_\mathrm{adv}$ | $0/255$                | $4/255$                | $8/255$                | $16/255$               | Optimal Accuracy       |
> > > |:------------------------------------:|:------------------------:|:------------------------:|:------------------------:|:------------------------:|:------------------------:|
> > > | None (Clean)                       | $94.59$                | $90.31$                | $84.88$                | $73.78$                | $94.59$                |
> > > | AdvPoison (original)                          | $9.91$  | $88.98$ | $83.11$ | $71.31$ | $88.98$ |
> > > | AdvPoison (Hybrid)                          | $\textbf{4.16}$  | $82.74$ | $78.65$ | $67.17$ | $82.74$ |
> > > | **Ours (Hybrid)**                           | $12.93$ | $\textbf{76.55}$ | $\textbf{74.30}$ | $\textbf{65.75}$ | $\textbf{76.55}$ |
> > >
> > > As can be seen, our hybrid technique is generally effective since it also substantially improves the original AdvPoison. Specifically, the performance of AdvPoison (Hybrid) is still much worse than Ours (Hybrid), confirming the superiority of our new loss function. We will include the above results in the final version of our paper.
> > >
> > > We would also like to mention that REM, ADVIN, and Unlearnable are essentially not good choices for implementing a hybrid attack since they are based on computationally expensive, bi-level iterative training rather than efficient, pre-training.
> > >
> > >
> > > > report the strongest defense against each attack in Table 1.
> > >
> > > Following our previous discussion, we have decided to focus Section 4 on challenging the consensus, i.e., showing that AT can be compromised under $\epsilon_\mathrm{poi}\leq\epsilon_\mathrm{adv}$, and Section 5 on a practical defense with an adjustable AT budget. Based on this structure, Table 1 in Section 4 still reports the results under $\epsilon_\mathrm{poi}=\epsilon_\mathrm{adv}$.
> > > In Section 5, we will first discuss our new observation about the independent roles of robust vs. non-robust features in poisoning AT vs. ST models. We will also show that this new observation indicates a general problem that all current attacks cannot achieve satisfactory performance when the AT budget of the defense is allowed to be adjusted (see the following table). Then, we address this problem by introducing a new hybrid technique and validating its general effectiveness based on the results in the above table.
> > >
> > > We believe the above way of presentation makes sense, and your suggestions are also welcome if you think there is a better way.
> > >
> > > | Method ($\epsilon_\mathrm{poi}=8/255$) $\backslash$ $\epsilon_\mathrm{adv}$ | $0/255$                | $4/255$                | $8/255$                | $16/255$               | Optimal Accuracy       |
> > > |:------------------------------------:|:------------------------:|:------------------------:|:------------------------:|:------------------------:|:------------------------:|
> > > | None (Clean)                       | $94.59$                | $90.31$                | $84.88$                | $73.78$                | $94.59$                |
> > > | AdvPoison                          | $\textbf{9.91}$  | $88.98$ | $83.11$ | $71.31$ | $88.98$ |
> > > | REM                          | $25.59$ | $\textbf{46.57}$ | $84.21$ | $85.76$ | $85.76$ |
> > > | ADVIN                         | $77.31$   | $90.08$ | $86.76$ | $72.16$ | $90.08$ |
> > > | Unlearnable                         | $25.69$   | $90.47$ | $84.91$ | $79.81$ | $90.47$ |
> > > | Hypocritical                       | $74.06$   | $91.18$ | $84.96$ | $73.33$ | $91.18$ |
> > > | Hypocritical+                     | $75.22$   | $84.82$ | $86.56$ | $82.26$ | $86.56$ |
> > > | **Ours**                           | $83.10$ | $75.39$ | $\textbf{71.51}$ | $\textbf{63.73}$ | $\textbf{83.10}$ |

---

### Author Response · Authors · 2022-11-08
**Common response to all reviewers**

We thank all reviewers for their helpful comments. The reviewers recognize the merits of our paper in various aspects, such as technical novelty and good writing (all reviewers), extensive evaluations (Reviewer VPK4, Reviewer VP8n, Reviewer ffZU), and impressive results (Reviewer VPK4, Reviewer VP8n, Reviewer ffZU).

We respectfully point out that the claim of Reviewer DCRE that leads to their “reject” score is actually based on a **factual error**. Specifically, after fixing an obvious implementation error in their provided code, their claim does not hold anymore.

In the following, we address all reviewers' comments. We also upload a revised version of our paper with all changes marked in blue.

---

### Decision · Program_Chairs · 2023-01-20

**Decision:**

Accept: notable-top-25%

**Justification For Why Not Higher Score:**

Concerns of Reviewer VPK4, overclaiming in some places as highlighted by DCRE

**Justification For Why Not Lower Score:**

A strong paper that challenges conventional wisdom regarding adversarial training and robustness

**Metareview: Summary, Strengths And Weaknesses:**

This paper received significant discussion from reviewers. They were pleased to see a clean-label attack that appears relatively powerful against adversarial training. There were some initial concerns (e.g., whether the attack is effective when there is no defense), but these seemed to be mostly assuaged by the authors responses and follow-up experiments. One critique was that it felt questionable about knowledge of the values of eps from the defender (though I think this was addressed in rebuttal), see also, the other concerns of Reviewer VPK4. The authors are expected to update the final version of the paper with all the changes suggested by the reviewers, especially DCRE who felt some of the claims needed to be toned down.

**Note From Pc:**

if the above contains the word "oral" or "spotlight" please see: "oral" presentation means -> notable-top-5% and "spotlight" means -> notable-top-25%. As stated in our emails, we are disassociating presentation type from AC recommendations